# Channeling the Force: Piezo1 Mechanotransduction in Cancer Metastasis

**DOI:** 10.3390/cells10112815

**Published:** 2021-10-20

**Authors:** Jenna A. Dombroski, Jacob M. Hope, Nicole S. Sarna, Michael R. King

**Affiliations:** King Lab, Department of Biomedical Engineering, Vanderbilt University, 5824 Stevenson Center, Nashville, TN 37235, USA; jenna.dombroski@vanderbilt.edu (J.A.D.); jacob.m.hope@vanderbilt.edu (J.M.H.); nicole.s.sarna@vanderbilt.edu (N.S.S.)

**Keywords:** Piezo1, mechanotransduction, cancer metastasis

## Abstract

Cancer metastasis is one of the leading causes of death worldwide, motivating research into identifying new methods of preventing cancer metastasis. Recently there has been increasing interest in understanding how cancer cells transduce mechanical forces into biochemical signals, as metastasis is a process that consists of a wide range of physical forces. For instance, the circulatory system through which disseminating cancer cells must transit is an environment characterized by variable fluid shear stress due to blood flow. Cancer cells and other cells can transduce physical stimuli into biochemical responses using the mechanosensitive ion channel Piezo1, which is activated by membrane deformations that occur when cells are exposed to physical forces. When active, Piezo1 opens, allowing for calcium flux into the cell. Calcium, as a ubiquitous second-messenger cation, is associated with many signaling pathways involved in cancer metastasis, such as angiogenesis, cell migration, intravasation, and proliferation. In this review, we discuss the roles of Piezo1 in each stage of cancer metastasis in addition to its roles in immune cell activation and cancer cell death.

## 1. Introduction

Cancer metastasis accounts for 90% of cancer-related deaths in patients [1]. Metastasis consists of five major steps: (1) invasion and migration, (2) intravasation, (3) dissemination, (4) extravasation, and (5) colonization [2] (Figure 1). Despite the high morbidity associated with cancer metastasis, there are many physiological barriers that cancer cells must overcome to successfully metastasize. Cancer cells must undergo transformation to develop increased motility and to allow for degradation of the surrounding extracellular matrix (ECM) to find and create pathways to escape the primary tumor [3]. Cancer cells can escape via squeezing through the endothelial cell wall or by promoting the growth of leaky vasculature [4]. In circulation, cancer cells are exposed to elevated fluid shear stresses that can cause damage to the cells and lead to cell death [5,6]. Thus, cells must develop mechanisms of resisting the damage by these physical forces [7]. When exiting the circulatory system, cancer cells must tether to the endothelium, pass through the endothelial wall, and begin invading the distant site [8]. The distant organ must have a suitable environment for cancer cell survival and proliferation for effective colonization [9]. Throughout the entire process, the cancer cells must avoid immune detection or develop resistance to cytotoxic proteins expressed and secreted by immune cells [10]. In the clinic, cancer cells must also contend with chemotherapies [11,12]. Identifying the mechanisms and proteins that allow cancer cells to accomplish these steps in the metastatic process is paramount, as this understanding will determine new targets for anti-metastatic therapy.

Piezo1 is a mechanosensitive ion channel (MSC) that has recently been associated with multiple steps in the metastatic cascade (for a review on Piezo1 in different cancer types see De Felice and Alaimo [13]). The protein Piezo1 is activated through cell membrane deformations [14]. These deformations can be caused by mechanical forces, such as osmotic pressure, fluid shear stress, substrate stiffness, and confinement [15]. When Piezo1 is activated, it results in calcium influx, transducing mechanical forces into biochemical responses [16]. Calcium is a ubiquitous second-messenger that is responsible for a diverse host of cellular functions that are necessary for successful cancer metastasis. These functions include survival, metabolism, proliferation, migration, and cytoskeletal remodeling [17]. In this review paper, we will discuss the various roles of Piezo1 in cancer metastasis and how Piezo1 could be leveraged in future cancer therapies [15,16,18,19,20,21,22].

## 2. Piezo1 Structure

The specific structure of human Piezo1 has yet to be thoroughly investigated. However, mouse Piezo1 (mPiezo1) has been studied using cryo-electron microscopy [14,23,24,25] (for a detailed review of Piezo1 structure see Fang et al. [26]). The full-length Piezo1 molecule contains 2547 amino acids and forms a homotrimeric structure [25]. The complex consists of three propeller-like blades on the extracellular region of the membrane, a single extracellular cap, an intracellular anchor region, three intracellular beam regions, and finally, a pore conducting path [26] (Figure 2). Piezo1 transduces mechanical signals, such as shear stress and osmotic pressure, into biochemical responses by sensing deformations in the cell membrane local to the channel. The blades of Piezo1 sense the deformations in the cell membrane, creating a force that is transmitted to the intracellular beams. The beams act to amplify the force felt by the blades as they operate through a lever-like apparatus that is positioned closer to the pore than to the blades along the beam. After the blades sense membrane deformation, the beams pull on the central pore, opening the cap and allowing for calcium ion flux [14,23].

## 3. Angiogenesis

In cancer, angiogenesis is the growth of blood vessels around the tumor [27]. New blood vessels support tumor growth by supplying oxygen and nutrients, and aid in the metastatic process by providing a means of egress for cancer cells from the primary tumor [27,28]. This escape of cancer cells during angiogenesis is facilitated by leaky vasculature formation [29]. Angiogenesis is promoted by the signaling protein vascular endothelial growth factor (VEGF), which targets tumor endothelial cells and is often upregulated in tumors [30]. Piezo1 promotes the expression of HIF-1α, which promotes VEGF as a downstream target [21,31,32]. HIF-1α and VEGF expression have been observed to be inhibited by the silencing of Piezo1 [21]. In one study, Piezo1 promoted VEGF expression in colon cancer, with Piezo1 expression elevated in colon cancer tissue and acting as a prognostic factor for patients [21]. However, within the same study, the overexpression of Piezo1 reduced VEGF expression, further implicating the dynamic role of Piezo1 in VEGF expression and cancer metastasis [21].

Endothelial cell migration is an important step in angiogenesis [33]. Studies have shown that Piezo1 knockdown prevented human umbilical vein endothelial cells (HUVECs) from migrating or aligning with VEGF [34]. Piezo1 causes HUVECs to align and grow with fluid shear stress, a force which is present in the vasculature [26,34,35]. This process is supported by Piezo1-activating calpains, causing the turnover of focal adhesions and the remodeling of the cytoskeleton [36,37]. Piezo1 senses laminar shear stress from blood flow resulting in focal adhesion regulation and endothelial cell alignment via activation of the membrane-type 1 matrix metalloproteinase (MT1-MMP) enzyme [38]. This process mediates angiogenesis since MT1-MMP functions by degrading the fibrin matrix to allow for vasculature growth [39]. The sarcoplasmic/endoplasmic-reticulum Ca^2+^ ATPase (SERCA) 2 enzyme, which maintains calcium ion homeostasis in cells, is a binding protein of Piezo1 and important to this alignment process during vascularization [40]. SERCA2-mediated inhibition of Piezo1 suppresses the migration of endothelial cells [40].

While Piezo1 has not always been explored in the context of cancer, its role in blood vessel growth is evident. This role may prompt future studies of Piezo1 inhibition as a potential angiogenesis-targeting therapy.

## 4. Invasion and Migration

Before tumors can progress in the metastatic cascade and enter the bloodstream, cancer cells undergo conformational changes and alter the surrounding microenvironment [41]. Piezo1 activation promotes migration in breast, gastric, colorectal, pancreatic, and prostate cancer cells [20,21,42,43,44]. In breast cancer, Piezo1 initiates Akt/mTOR signaling, a pathway responsible for regulating cell motility and survival [45]. Piezo1 is a binding protein to trefoil factor family 1 (TFF1), contributing to invasion and migration for gastric cancer cells by reducing anoikis, increasing motility, and promoting migration [42,46].

Epithelial-to-mesenchymal transition (EMT), a process by which epithelial cells undergo changes to become more mesenchymal in phenotype, is promoted by Piezo1 [21,32,47,48]. This process enhances migration, invasive potential, and resistance to cell death, leading to increased cancer metastasis [47]. Activation of Piezo1 results in translocation of Yes-associated protein 1 (YAP1) to the nucleus, thereby increasing cell motility in a variety of cell types [48,49]. In a study with cholangiocarcinoma cells, Piezo1 knockout inhibited EMT in cancer cells [48]. As mentioned previously, Piezo1 enhances the expression of HIF-1α, a significant promoter of angiogenesis [21,32]. Besides vascular growth, HIF-1α can also increase EMT [21,32]. EMT can also be promoted by increased matrix stiffness, which, in a study of glioblastoma, Piezo1 was able to induce [18,50,51]. Piezo1 induces tissue stiffening through the binding of integrins and focal adhesions to stiffer matrixes. The increased binding of integrins and focal adhesions then increases cell membrane tension, allowing for increased Piezo1 activity. This increased Piezo1 activation by matrix stiffness promoted the upregulation of mRNA of proteins associated with matrix remodeling. The explicit cell signaling proteins for this increased mRNA expression were not identified, however [18].

Calpains activated by Piezo1 increase cancer cell motility through reorganization of actin and remodeling of the cytoskeleton [36,37,52]. Motility is also enhanced through the mechanosensitive response of Piezo1 to confinement, which is upregulated in cancers [53]. As part of this process, the influx of calcium suppresses cAMP-dependent protein kinase A (PKA) through PDE1 activation, thus directing cell locomotion [53]. PKA has many functions as a tumor suppressor [53,54,55,56]. With its ability to sense local mechanical cues, Piezo1 causes calcium “flickers”, which have been shown to cause cells to migrate in a specific direction via Ca^2+^-dependent proteins [57,58].

Piezo1 has also been shown to affect migration in breast cancer [20,59]. In a study comparing MCF-7 breast cancer cells to neoplastic MCF-10A healthy mammary glands, which lack a mechanoactivated current, MCF-7 cells expressed significantly higher levels of exogenous Piezo1 mRNA [20]. Following inhibition with GsMTx4, an inhibitor of MSCs, motility and velocity of MCF-7 cells were inhibited, while MCF-10A cells were not affected [20,60]. Furthermore, in this study, Piezo1 was associated with shorter survival times for breast cancer patients [20]. Piezo1 also plays a significant role in gastric cancer. Piezo1 knockdown has been observed to decrease migration capacity in gastric cancer cells during in vitro assays through the downregulation of the β1 subunit of integrin [42]. These effects are attributed to the interaction between Piezo1 and the TFF1 protein, which contributes to invasion and migration [42]. In synovial sarcomas, Piezo1 is upregulated [19]. In one study of the SW982 synovial sarcoma cell line, Piezo1 knockdown resulted in decreased cell migration [19].

The promotion of migration by Piezo1 is not well understood in all settings, as in lung cancer where Piezo1 is often downregulated [61,62,63]. Piezo1 downregulation can accelerate cell migration in lung cancer by promoting ameboid-type migration [61,62]. In MDA-MB-231 cells, Piezo1 inhibited blebbing, providing more evidence that Piezo1 prevents ameboid migration [64]. In studies of non-small cell lung cancer (NSCLC), Piezo1 knockdown enhanced migration and tumor growth [63].

## 5. Intravasation and Extravasation

Although occurring at different stages in the metastatic cascade, intravasation and extravasation play similar roles in the process. As previously described, Piezo1 plays a role in promoting angiogenesis, which allows for the escape of cancer cells through leaky vasculature. This escape facilitates cancer cell intravasation into blood flow [21,29,31,32]. Piezo1 has been observed to increase endothelial cell permeability in lung cancer through calpain activity by reducing tight junctions [16]. Matrix stiffness is also promoted by Piezo1 expression [18,65]. Enhanced matrix stiffness promotes N-cadherin expression on endothelial cells, which supports a mesenchymal phenotype in cancer cells, allowing them to squeeze between cells during intravasation [38,66]. In glioma cells, Piezo1 is localized at focal adhesions, reinforcing tissue stiffening, and further upregulating Piezo1. This feedforward system promotes malignant progression [18].

Piezo1 has also been found to play a role in shear stress-induced release of ATP in red blood cells (RBCs), where paracrine signaling prompts the formation of inter-endothelial junctions and supports intravasation and extravasation of cancer cells [67].

Fluid shear stress has played a role in cell extravasation in vitro, which could extend to the importance of Piezo1 in this process [68,69]. In one study, polymorphonuclear neutrophil (PMN)-facilitated melanoma cell extravasation was enhanced under flow conditions [68]. In another study, hydrodynamic shear stress via transendothelial assay enhanced breast tumor cell extravasation [69]. Although Piezo1 is not specifically mentioned, the impact of Ca^2+^ influx, an important result of Piezo1 opening, has been observed in cancer cell extravasation [70]. In one study, Ca^2+^ inhibition resulted in suppressed EGF-migration and prevented nasopharyngeal carcinoma cell extravasation, demonstrating the importance of Ca^2+^ influx [70].

Other MSCs have been investigated for their role in cell intravasation and extravasation, and although they do not always function in the same manner as Piezo1, it reinforces the importance of MSCs to these processes. Transient receptor potential melastatin 7 (TRPM7) is a fluid shear stress sensor that has reduced activity when cancer cells are undergoing intravasation [71]. Transient receptor potential vanilloid 4 (TRPV4) is another MSC that plays a significant role in intravasation and extravasation by promoting EMT, which facilitates intravasation [72]. TRPV4 has been observed to be upregulated in metastatic breast cancer cell lines 4T07, 4T1, and MDA-MB-468, compared to primary tumors that were unable to undergo extravasation [73,74]. Upregulation of TRPV4 resulted in increased deformability and migratory ability in these cells, facilitating both intravasation and extravasation [72,74]. This increased deformability promotes morphological softening, which allows cells to exit the vasculature more easily during extravasation [75]. Additionally, breast cancer cells and extravasating leukocytes have similar expression of specific phosphorylated proteins necessary for actin cytoskeletal remodeling and deformability, which suggests the importance of TRPV4 for the phosphorylation of these proteins in extravasation [76,77,78]. The MSC purinergic type 2 X7 receptor (P2X7) also plays a significant role in intravasation and extravasation [72]. P2X7 supports EMT, which aids in intravasation [79,80]. P2X7 activation also promotes MMP activity, thereby facilitating intravasation, in MDA-MB-435 cells [79,81,82]. This significance is further supported by cancer cells without MMPs present being unable to intravasate [83]. Extravasation is supported by MMP-facilitated degradation of the ECM in MDA-MB-435 cells [79,81]. These findings with other MSCs suggest that the role of Piezo1 in intravasation and extravasation is more complicated than its role in other aspects of metastasis.

## 6. Dissemination

Many studies have indicated the significance of EMT in the early stages of cancer cell dissemination during metastasis [84,85]. In one study of melanoma cells, mesenchymal-related genes N-cadherin and E-cadherin had significantly reduced expression after Piezo1 knockdown [86]. This reduction in mesenchymal-promoting genes supports the role that Piezo1 plays in EMT and, therefore, cancer cell dissemination. As previously mentioned, Piezo1 also promotes EMT through tissue stiffening [18,50,51]. In one study of RasV12-transformed cells, Piezo1 was found to be essential for dissemination [87]. While these studies were performed with *Drosophila* intestinal stem cells (ISCs) and enteroblasts (EBs), the findings can easily extend to disseminating cancer cells due to the significance of Ras in cancer. Ras proteins are binary switches that are vital to signal transduction and are often mutated in different types of cancer [88,89,90,91]. Additionally, higher hazard ratios and shorter survival times were observed in breast cancer patients with primary tumors exhibiting high Piezo1 mRNA levels, indicating the role of Piezo1 in metastatic dissemination [20].

As previously mentioned, Piezo1 is a binding protein to TFF1, which has been observed to impact different aspects of cancer metastasis. In several studies, enhanced levels of TFF1 were associated with the promotion of dissemination in different cancers, including those originating in the breast and GI system [42,92,93]. Despite these findings, TFF1 plays a more complicated role in dissemination. For instance, it has been observed to inhibit EMT in pancreatic intraepithelial neoplasm (PanIN) [94]. It has also acted as a tumor suppressor in stomach cancers and is not expressed in highly metastatic breast cancer cell lines such as MDA-MB-231 cells [95].

Dissemination of circulating tumor cells (CTCs) is supported by platelets, which bind to CTCs during circulation and protect them from the harsh forces of the circulatory system [96,97,98]. In circulation, CTCs induce platelet activation and aggregation, becoming enveloped with platelets to shield them not only from fluid shear stress in blood flow but from the immune system and TNF-α-mediated cytotoxicity [99]. Piezo1 facilitates this binding by opening in response to blood flow forces, allowing a calcium influx to activate platelets [100,101]. Piezo1 also aids RBCs in their transit through circulation by allowing them to deform to fit through tight capillaries [102,103,104]. It is possible that these changes could occur in CTCs as well, making them better suited for survival in the circulatory system. In one study, breast cancer cells were able to survive in circulation as well as immune cells, which could suggest the importance of Piezo1 in this process [5].

The role of Piezo1 has been explored in a variety of different types of disseminating cells. While its connection to CTCs has been less frequently explored beyond other cells, such as platelets, assisting in CTC trafficking, it appears to play a significant role in this process nevertheless.

## 7. Piezo1 in Colonization

Piezo1 has not been directly linked to the colonization of distant tumor sites following cancer cell dissemination. However, increased intraosseous pressure was previously shown to support successful colonization of the bone by prostate cancer cells [105]. Piezo1 has also been shown to be activated by increased mechanical loading of bones, suggesting a possible link between Piezo1 and metastatic colonization [106]. Piezo1 has been linked to the expression and activation of proteins associated with successful colonization and increased proliferation, such as HIF-1α, Akt/mTOR, and ERK [21,43,107].

Piezo1 has been shown to induce HIF-1α expression in gastric cancer cells through calcium influx in response to mechanical force. Additionally, it was shown that Piezo1 knockdown in gastric cancer cells significantly reduced HIF-1α expression and metastasis [108]. An additional study identified HIF-1α to be essential to metabolic reprogramming that promoted the formation of metastatic lesions. HIF-1α caused the cells to switch from mitochondrial oxidative phosphorylation to anerobic glycolysis. This process also reduced the intratumoral reactive oxygen species (ROS) to non-cytotoxic levels, allowing for successful metastatic tumor formation [109].

Akt/mTOR signaling has been found to promote cancer cell proliferation and is known to be activated in prostate cancer cells [43]. A study of Piezo1 in prostate cancer showed that shRNA knockdown of Piezo1 significantly reduced proliferation and caused cell-cycle arrest by reducing the activity of Akt/mTOR in a calcium-mediated process [43]. A breast cancer study identified Akt/mTOR as being a major component of successful metastatic colonization of the lungs. The authors demonstrated that increased laminar shear stress caused a significant upregulation of Akt/mTOR signaling [110]. The enhanced Akt/mTOR signaling was reliant on the expression of the oncogene caveolin-1, which was also upregulated by the laminar shear stress [110,111]. Caveolin-1 expression has been tied to calcium signaling since its expression depends on the activation of the calcium-reliant transcription factor, NFAT [112]. This finding suggests a possible link between Piezo1, caveolin-1, and Akt/mTOR, as Piezo1 activation is induced by laminar shear stress and causes calcium influx [113].

ERK1/2 was shown to be a phosphorylation target downstream of calcium influx by Piezo1 activation in canine epithelial cells [15]. The phosphorylated ERK1/2 then caused a significant increase in the mitosis of the cells. ERK1/2 has been linked to cancer progression and successful colony formation [114,115] In one study, Piezo1 was activated by stretching the substrate on which the cells were cultured, causing deformation of the cell membranes. The stretching of the cells induced a five-fold increase in proliferation [15]. While ERK1/2 and Piezo1 have not yet been linked to colony formation in cancer metastasis, this work does suggest a further link between the two, as bone tissue is exposed to different loading conditions in vivo that can give rise to different stretch patterns capable of inducing this rapid cell division in metastatic cancer cells that have invaded the bone [116].

## 8. Piezo1 Survival and Apoptosis

The roles of Piezo1 in both survival and apoptosis further demonstrate the diverse roles of Piezo1 in cancer metastasis. For example, in a previous study of colon cancer, Piezo1 knockdown significantly reduced the viability of HCT116 and SW480 cells. However, increasing the activation of Piezo1 through Yoda1 reduced cell viability by causing mitochondrial depolarization [21]. A study of Piezo1 and cell viability in synovial sarcoma showed that reducing the expression of Piezo1 caused a significant reduction in cell viability as well. Increasing Piezo1 activation using the Yoda1 agonist, however, did not have any effect on cell viability [19]. The authors of this study hypothesized that Piezo1 may not play a direct role in cell survival or death. Instead, they suggested that Piezo1 may be important in cell adhesion as was demonstrated in previous work [19,117]. This finding suggests that Piezo1 may prevent cell death by promoting cell adhesion, thus preventing anoikis [118]. However, this is contradicted by another study in which Piezo1 promoted extrusion of overcrowded epithelial cells that resulted in cell death [119]. When Piezo1 was knocked down, however, the epithelial cells continued to overgrow.

The extrusion-related cell death, however, may be of less importance in cancer cells because anoikis is induced through intrinsic apoptosis [120]. Intrinsic apoptosis involves mitochondrial dysfunction that results in the release of proapoptotic proteins, such as cytochrome c and Smac, from the mitochondria [121]. Intrinsic apoptosis is inhibited by Bcl-2, which is commonly upregulated in metastatic cancer cells, suggesting the extrusion-related cell death by Piezo1 is less likely to occur in metastatic cancer cells [122].

Piezo1 has also been linked to increased cancer cell survival in gastric and prostate cancer [43,108]. In gastric cancer, Piezo1 inhibition through knockdown reduced cancer cell growth and increased the sensitivity of the cells to chemotherapy [108,123]. In prostate cancer cells, a similar effect was seen where Piezo1 knockdown caused a small reduction in cell viability in vitro in DU145 cells. However, the Piezo1 knockdown in vivo seemed to have a more prominent impact on tumor growth [43].

Piezo1 has also been linked to apoptosis in cancer cells and other cell types primarily through calpain-mediated apoptosis [22,124,125]. Calpains are proteases that are activated by calcium influx and can induce apoptosis by cleaving Bid to its active form of truncated Bid (tBid) [126]. Calpains can also cleave the Bax/Bak inhibitor protein Bcl-2 [127]. These activities by calpains allow for intrinsic apoptosis to occur and lead to caspase-mediated cell death [128]. In prostate cancer cells, Piezo1 activation by fluid shear stress and Yoda1 significantly increased the sensitivity of the cells to TRAIL-mediated apoptosis. This sensitization by Piezo1 activation was reliant on calcium influx, calpain activation, and increased mitochondrial depolarization [22]. These findings are further supported by another study that used MDA-MB-231 breast cancer cells. This study demonstrated that cyclic stretching of the cancer cells caused a Piezo1-mediated calcium influx that resulted in Bax activation and intrinsic apoptosis [129]. Interestingly, the authors also found that the cyclic stretch promoted cell growth of healthy cells.

## 9. Piezo1 in Immune Cells

Calcium signaling in immune cells is known to be crucial to their physiological behavior and expression of enzymes and transcription factors that regulate inflammatory function [130,131]. Additionally, the essential role of mechanical forces in directing immune cell function has received increasing attention, as many disease states involve increased tissue stiffness [132,133,134]. While the exact mechanisms through which mechanosensitive ion channels regulate the immune system are uncertain, the role of Piezo1 in leukocytes has gained recent attention. The growing interest in Piezo1 in immune cells may provide insight into how it functions in cancer cells and how modulating Piezo1 activity may lead to potential cancer immunotherapies and treatment of infectious disease.

Physical cues and external stimuli in the microenvironment drive a cell’s physiological function and lead to numerous downstream signaling effects. Piezo1 has been identified as a critical sensor of mechanical stress in numerous myeloid and lymphoid cells, indicating its role in the immune system by linking mechanical forces with immune regulation [135,136,137,138]. In one study, Piezo1 was determined as the primary sensor of physical forces in myeloid cells, as Piezo1-deficient leukocytes were unable to transduce mechanical signals [135]. They also demonstrated in vivo that deletion of Piezo1 on myeloid cells diminished the progression of pancreatic cancer and protected against polymicrobial sepsis [135]. In another study, it was reported that the pro-inflammatory response of myeloid cells was entirely dependent on the mechanosensation of Piezo1 when exposed to cyclical pressure [139]. Mechanistically, the authors determined that Piezo1 signaling in myeloid cells leads to activation of activator protein-1 (AP-1), transcription of endothelin-1 (Edn-1), and stabilization of H1F-1α [139].

Immune cell activation relies on chemical and physical cues to potentiate an appropriate immune response. As a mechanosensitive ion channel, Piezo1 has exhibited an essential role in immune cell activation. In one study, Piezo1 was determined to be a critical protein involved in human T cell activation [137]. By evaluating phosphorylation of ZAP70 and induction of CD69 transcripts—which are indicative of T cell activation—the Piezo1 agonist Yoda1 induced optimal T cell activation [137]. The proposed mechanism involves cell membrane stretch causing calcium influx through Piezo1, which activates calpain and triggers the reorganization of the cortical actin scaffold, thereby optimizing T cell receptor (TCR) signaling [137]. It has also been shown that Piezo1 is essential for the activation of macrophages [136]. This study’s results indicate that the activation of Piezo1 in macrophages is responsible for enhanced inflammation and decreased healing response, through increased nuclear factor kappa-light-chain-enhancer of activated B cells (NFκB) and decreased signal transducer and activator of transcription 6 (STAT6) activation, respectively [136]. Similarly, another study demonstrated the necessity of Piezo1 in monocytes in activating a pro-inflammatory response, which is essential for immunity against bacterial infection in the lungs [139]. However, the authors also reported exacerbated autoinflammation in a model of pulmonary fibrosis through the activation of Piezo1 [139]. Finally, Piezo1 has been shown to play an important role in dendritic cell (DC) activation and function when exposed to mechanical stress [138]. In this study, Yoda1 strongly upregulated the production of pro-inflammatory cytokines IL-6 and TNF- α [138]. The authors also found that Piezo1-deficient DCs mildly suppressed the DC-mediated anti-tumor response in a syngeneic ovalbumin (OVA) mouse model [138].

Piezo1 has exhibited essential roles in cell physiology as a transducer of physical stimuli and potentiator of chemical signaling. The demonstrated importance of Piezo1 activity in immune cells gives rise to the potential of using Piezo1 to modulate cancer immunotherapies and infectious disease treatment. Additionally, a thorough understanding of the effects of Pizeo1 in immune cell activation may give rise to high-throughput immune cell activation for adoptive cell therapies. However, future studies will be necessary to evaluate the effects of Pizeo1 activity in immune cells on disease and cancer progression.

## 10. Conclusions

The diverse responses of Piezo1 in inducing cell death, increasing cell survival, modulating immune cell activation, and enhancing cancer metastasis demonstrates the complexity of cancer progression (Table 1). It suggests that both inhibiting and activating Piezo1 may have potential in treating the disease. However, observations to date highlight the individuality of each cancer case. Different organs have diverse environments with various signaling proteins and mechanical forces at play. In one context, Piezo1 may be an ideal target for inhibition to prevent metastasis, but in another, it may be better to activate Piezo1 to prevent metastasis. Piezo1 encompasses the broad range of effects that mechanical forces can have on cancer progression and indicates the clear need for further research into the effects of mechanotransduction in cancer metastasis.

## Figures and Tables

**Figure 1 cells-10-02815-f001:**
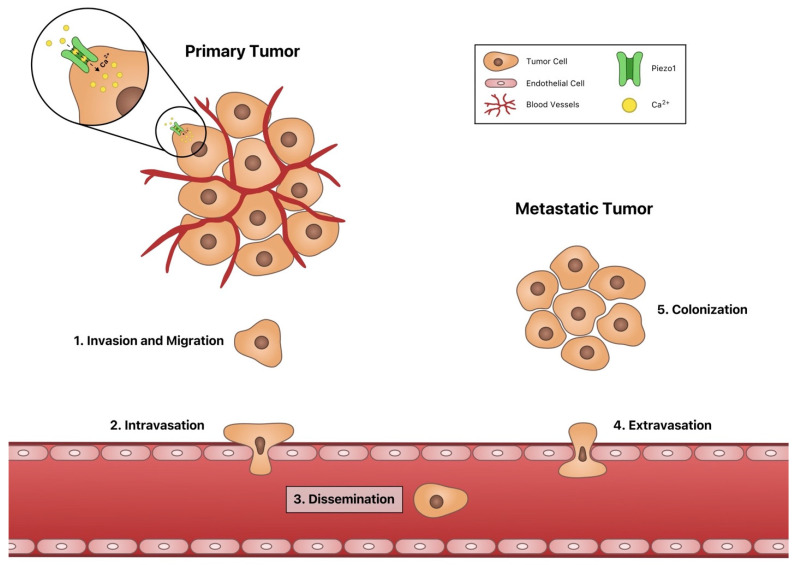
Schematic of cancer metastasis and the five major steps of metastasis.

**Figure 2 cells-10-02815-f002:**
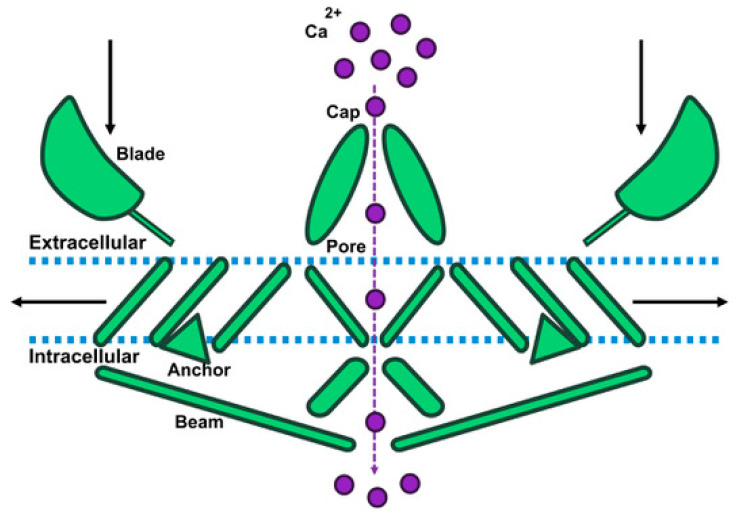
Schematic of Piezo1 and pore channel opening.

**Table 1 cells-10-02815-t001:** Cancers where Piezo1 either promoted or inhibited the indicated metastatic “stage”.

Metastatic Stage	Promoted Cancers	Inhibited Cancers
Angiogenesis	Colon [21]	
Invasion and migration	Breast [20], Colon [21], Gastric [42], Pancreatic [43], Prostate [44], Liver [47], Glioblastoma [18], Synovial [19]	Lung [62]
Intravasation and Extravasation	Lung [16]	
Dissemination	Melanoma [85]	
Colonization	No direct links	No direct links
Apoptosis	Colon [21], Synovial [19], Breast [129]	Colon [21], Gastric [107], Prostate [43]

“Promoted cancers” indicates cancer types that have been studied where Piezo1 promotes the indicated metastatic stage; “Inhibited cancers” indicates cancer types that have been studied where Piezo1 inhibits the indicated metastatic stage.

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
