# Peer review of "Channeling the Force: Piezo1 Mechanotransduction in Cancer Metastasis"

_cells, 2021, doi:10.3390/cells10112815_

Round 1

Reviewer 1 Report

The manuscript presented reviews the role of the Piezo1 channel in various aspects of how cancer cells transduce mechanical signals and the successive impact on potential metastasis: angiogenesis, cell migration, intravasation and proliferation. 

The review is well organized in sequential sections and very well written, the background literature is also extensively reviewed and incorporated. While there are some confounding reports on the piezo1 noted in various cancers, the manuscript would benefit significantly if these can be noted/summarized in a table, sorted by cancer type and then by piezo1 and associated findings. (These literature reports may not lend themselves to such tabulation, but if at all possible, that would make the information easier to digest).

Although mentioned multiple times in the manuscript, that Piezo1 influences/promotes matrix stiffening, it is not clear if the precise mechanism or model for this is known: whether this is due to cell density/proliferative changes or matrix remodeling; or combination thereof and if so which contributes primarily? Any discussion of this from the available literature would be very valuable to this manuscript.

Minor comments:

  1. The TRPV4 sentence on line 174-75 on page 5 is missing a word/phrase.

Author Response

We would like to thank the reviewers for their time and effort in reviewing our manuscript. We have strived to address all of their comments to increase the impact and readability of the manuscript.

Reviewer 1:

Table: We added a table that highlights the prometastatic and antimetastatic roles of Piezo1 in different cancers. We also added a reference to another review article that discusses Piezo1 in specific cancer types that may aid in understanding the nuances of Piezo1 in different cancers.

Piezo1 promotes matrix stiffness: The role of Piezo1 in influencing matrix stiffness is not yet fully established. However, we did discuss in greater detail the glioblastoma study that first identified this phenomenon in the Invasion and migration section at lines 118 – 125. The discussion reads, “EMT can also be promoted by increased matrix stiffness, which, in a study of glioblastoma, Piezo1 was able to induce tissue stiffening18,49,50. Piezo1 induces tissue stiffening through the binding of integrins and focal adhesions to stiffer matrixes. The increased binding of integrins and focal adhesions then increases cell membrane tension, allowing for increased Piezo1 activity. This increased Piezo1 activation by matrix stiffness promoted the upregulation of mRNA of proteins associated with matrix remodeling. The explicit cell signaling proteins for this increased mRNA expression were not identified, however18.”

Minor Comment 1: We combined the sentences on lines 174 – 176 to better connect the statements and links between TRPV4 and intravasation. The lines now read, “Transient receptor potential vanilloid 4 (TRPV4) is another MSC that plays a significant role in intravasation and extravasation by promoting EMT, which facilitates intravasation70.”

Reviewer 2 Report

This manuscript by Dombroski JA et al. is a timely review article. Recently, De Felice S and Alaimo A published a similar review article on Piezo channels in cancer in ‘Cancers’, a MDPI journal. They focused on the roles of Piezo channels in different cancer types and Piezo-related intracellular signaling in cancer. On the other hand, this manuscript by Dombroski JA et al focused on the role of Piezo channels in cancer metastasis. The paper is well-written and the chapter compositions and figures are well-organized. I have one comment.

1. To enhance the originality of this review article, the authors should remove/amend/modulate the topics described by De Felice and Alaimo in ‘Cancers’ (2020).

Author Response

We would like to thank the reviewers for their time and effort in reviewing our manuscript. We have strived to address all of their comments to increase the impact and readability of the manuscript.

Reviewer 2:

Comment 1: We believe that our article is original and distinct from De Felice & Alaimo’s article. While the two articles do have significant overlap in disease state, we approach cancer from two different directions. The work of De Felice & Alaimo discusses the roles of Piezo channels more wholistically in different cancer types, whereas our article focuses on the roles of Piezo1 within cancer metastasis. There is overlap in that we mention in which cancer types Piezo1 promotes or hinders cancer metastasis. We feel if we were to not mention this information it could detract from the impact of our article, as Piezo1 has been shown to have divergent roles in different cancer types.

To credit De Felice & Alaimo we included a parenthetical statement in the introduction where Piezo1 in cancer is first introduced at lines 46 – 47. The parenthetical statement reads “(for a review on Piezo1 in different cancer types see De Felice & Alaimo13).